# SAC: Accelerating and Structuring Self-Attention via Sparse Adaptive Connection

**Xiaoya Li*♣, Yuxian Meng*♣, Mingxin Zhou♣, Qinghong Han♣, Fei Wu♠ and Jiwei Li ♠♣**
♠ Computer Science Department, Zhejiang University
♣ Shannon.AI
{xiaoya_li,yuxian_meng,mingxin_zhou,qinghong_han,jiwei_li}@shannonai.com

## Abstract

While the self-attention mechanism has been widely used in a wide variety of tasks, it has the unfortunate property of a quadratic cost with respect to the input length, which makes it difficult to deal with long inputs. In this paper, we present a method for accelerating and structuring self-attentions: Sparse Adaptive Connection (SAC). In SAC, we regard the input sequence as a graph and attention operations are performed between linked nodes. In contrast with previous self-attention models with pre-defined structures (edges), the model learns to construct attention edges to improve task-specific performances. In this way, the model is able to select the most salient nodes and reduce the quadratic complexity regardless of the sequence length. Based on SAC, we show that previous variants of self-attention models are its special cases. Through extensive experiments on neural machine translation, language modeling, graph representation learning and image classification, we demonstrate SAC is competitive with state-of-the-art models while significantly reducing memory cost.

## 1 Introduction

The self-attention mechanism has proved to benefit a wide range of fields and tasks, such as natural language processing (Vaswani et al., 2017; Dai et al., 2019), computer vision (Xu et al., 2015; Jaderberg et al., 2015; Bello et al., 2019; Parmar et al., 2019), video classification (Wang et al., 2018) and graph modeling (Veličković et al., 2018; Lee et al., 2019). The attention mechanism allows the model to attend to different parts of the input and capture the most salient features, and provides with the ability to handle dependencies between elements across long distances.

Two conspicuous problems stand out with the self-attention mechanism: (1) the memory complexity is quadratic with respect to the input length, making it infeasible to handle long sequences; and (2) the self-attention operations are performed in a pre-defined structure (usually fully-connected), which not only is costly but also lacks for the ability to learn the optimal attention structure optimized for the performance in the downstream tasks (Child et al., 2019; Sukhbaatar et al., 2019). These observations indicate that the fully-connected structure for self-attention operations can be replaced by a more sparse one where only specific edges are constructed for attention operations.

In this paper, we present Sparse Adaptive Connection (SAC), which replaces the fully-connected structure in self-attention with a sparse graph-like structure adapted to different tasks. In SAC, we regard the input as a graph where a node could be a token in the sequence or an flattened feature map of an image, and an edge between a pair of nodes represents they can attend to each other. To select edges, we propose an Edge Predictor which utilizes an LSTM model to dynamically predict pairs of nodes that represent two ends of an edge. In contrast with previous self-attention models with pre-defined attention structures (edges), SAC learns to construct attention edges adaptively, which are optimized to improve task-specific performances using reinforcement learning models. We

evaluate the proposed model on four tasks: neural machine translation, language modeling, graph representation learning and image classification, and demonstrate that SAC learns to select the most salient nodes to attend to each other and is free from heavy memory cost while achieving strong performances.

## 2    Related Work

Many recent researches have focused on modifying the structure of self-attention to relieve the computation and memory burden. Transformer-XL (Dai et al., 2019) enables learning long-term dependency by introducing a segment-level recurrence mechanism and a novel relative position encoding scheme (Shaw et al., 2018). (Guo et al., 2019) proposed Star Transformer, a variant of Transformer which replaces the fully-connected structure in self-attention with a star-shaped topology, where all tokens are connected with a relay node. (Child et al., 2019; Kitaev et al., 2020) suggest sparsifying Transformer by focusing only on a fraction of attention connections. (Child et al., 2019) introduced sparse factorizations of the attention matrix, which scale as $O(n\sqrt[p]{n})$ with the sequence length, and a set of sparse attention kernels which efficiently compute subsets of the attention matrix. (Kitaev et al., 2020) proposed Reformer, a more efficient yet complicated method for computing self-attention, where the dot-product attention is replaced by one that uses a locality-sensitive hashing, reducing the complexity from $O(n^2)$ to $O(n\log n)$. (Ye et al., 2019) partitioned the sequence to different multi-scale spans and attended the context information from fine-grain to coarse-grain as the relative distance increases. All above works rely on manually designed structures. Our work is inspired by (Sukhbaatar et al., 2019), which takes a step forward and uses a novel adaptive self-attention mechanism that can learn its optimal attention span for each head, and (Correia et al., 2019) which proposed adaptively sparse Transformer. Different from Yang et al. (2018), we use an LSTM to predict attention links which gives us finer control of how sparse we want self-attention to be.

Graph neural networks (GNNs) are known at learning local contextual information by encoding attribute features (Kipf and Welling, 2016; Hamilton et al., 2017b), but they are not able to explicitly distinguish the most salient nodes from all its neighbors, neither can they directly attend to the nodes that are beyond one-hop away. Much work has investigated the effect of attention on GNNs (Veličković et al., 2018; Abu-El-Haija et al., 2018; Lee et al., 2018; Veličković et al., 2019). (Veličković et al., 2018) extended self-attention to graphs by enabling each node to attend over its neighbors, achieving state-of-the-art results for semi-supervised node classification tasks. But they simply applied self-attention over graphs to all neighbors of a node, which might be a problem when dealing with large and noisy graphs where only few neighbors need to be aggregated. (Ye and Ji, 2019) proposed Sparse Graph Attention Network which uses a binary gate to control whether each edge should be engaged. However, these works lack ability to aggregate long-range dependencies in graphs, and they only consider neighbors that are one hop away. Various methods have been proposed to tackle this issue (Ye et al., 2020; Zhang et al., 2020; Pei et al., 2020). Similar to graphs, (Bello et al., 2019) introduced a novel two-dimensional relative self-attention mechanism for images and augmented convolutional operators with this self-attention method, showing systematic improvements on both image classification and object detection tasks across a wide range of architectures.

## 3    Background: Self-Attention

Given a set of nodes[1] $\{e_1, \cdots, e_N\}$ as inputs, self-attention iteratively computes the representation of $e_i$ in the $l$-th layer by attending to all its neighbors $\mathcal{N}(e_i)$, which is defined as follows:

$$\tilde{\mathbf{h}}_i^l = \sum_{e_j \in \mathcal{N}(e_i)} \alpha_{ij} \mathbf{v}_j^{l-1}, \ \alpha_{ij} = \text{softmax}\left(\frac{(\mathbf{q}_i^{l-1})^\top \mathbf{k}_j^{l-1}}{\sqrt{d}}\right) \tag{1}$$
$$\text{and} \quad \mathbf{q}_i^{l-1} = \mathbf{W}^Q \mathbf{h}_i^{l-1}, \ \mathbf{k}_j^{l-1} = \mathbf{W}^K \mathbf{h}_j^{l-1}, \ \mathbf{v}_j^{l-1} = \mathbf{W}^V \mathbf{h}_j^{l-1}$$

where $d$ is the hidden dimension, $\mathbf{W}^Q, \mathbf{W}^K, \mathbf{W}^V$ are learnable parameters and $\mathbf{q}, \mathbf{k}, \mathbf{v}$ correspond to queries, keys and values, respectively. The multi-head mechanism linearly projects the queries, keys

and values multiple times with different learned linear projections, and then performs self-attention in parallel, after which the results are concatenated and again projected:

$$\mathbf{h}_i^l = \text{Concat}(\tilde{\mathbf{h}}_i^{l,1}, \cdots, \tilde{\mathbf{h}}_i^{l,m})\mathbf{W}^O \tag{2}$$

where the superscript $^{1,\cdots,m}$ denotes the head number, and $\mathbf{W}^O$ is learnable parameters. After $L$ iterations, we obtain the final representation for each node $\mathbf{h}_i^L$.

## 4  Sparse Adaptive Connection for Self-Attention

The key point in SAC is to use to an LSTM edge predictor to predict edges for self-attention operations between nodes, where a node could be a token in the sequence or an flattened feature map of an image. Self-attention operations are performed between linked nodes instead of in a fully-connected manner. The LSTM edge predictor is optimized to improve task-specific performances using reinforcement learning models.

### 4.1  LSTM Edge Predictor

In SAC, an edge predictor is used to construct edges between nodes for self-attention operations. Suppose that we are given a set of nodes $\{e_1, \cdots, e_N\}$ with no edge between any pair of nodes when initialization, our aim is to generate edges using this edge predictor, with the total number $\alpha N$ for each layer, where $\alpha$ is a hyperparameter deciding how many edges should be constructed for each node on average. The Edge Predictor uses an LSTM model as a backbone and sequentially predicts edges. The prediction of an edge is decoupled into the prediction of the original node and the destination node pair.

More formally, the input to Edge Predictor is a special token "[SOS]", and the model proceeds to predict the original node and destination node of all edges ($2\alpha N$ nodes in total) for the first layer, denoted by $\{y_1^1, y_2^1, \cdots, y_{2\alpha N}^1\}$, where the superscript denoted the index of the layer and the subscript denoted the index of the predicted node. At each time step, the input to the LSTM model is the representation $\mathbf{h}_{y_t}$ for the node that has just been predicted. Then it is combined with the previously constructed representation $\mathbf{g}_t$ to obtain $\mathbf{g}_{t+1}$ representing the current time-step using LSTMs, and $\mathbf{g}_{t+1}$ is used to predict the following node using the softmax function. The projection matrix before softmax $\mathbf{W}$ shares embeddings with node representations, where each column $\mathbf{w}_i$ is the vector representation for node $e_i$. The probability of predicting node $y_{t+1}$ given $\mathbf{g}_{t+1}$ is thus given by:

$$p(y_{t+1} = e_i) = \frac{\exp{(\mathbf{g}_{t+1}^\mathsf{T} \cdot \mathbf{w}_i)}}{\sum_j \exp{(\mathbf{g}_{t+1}^\mathsf{T} \cdot \mathbf{w}_j)}} \tag{3}$$

This process is repeated $2\alpha N$ times. After the end of $\alpha N$ edge predictions, we update the representation for each node based on self-attention as will be detailed in Section 4.1 for different tasks, and proceed to the next layer. For node predictions in the following layer, the initial input now becomes hidden state for the last time-step of the previous layer. The entire process is repeated $L$ times, where $L$ denotes the number of self-attention layers and the resulted nodes in layer $l$ are $\{y_1^l, y_2^l, \cdots, y_{2\alpha N}^l\}$. Compared to separately predicting edges for each node, this approach is more flexible and gives us finer control of the total number of edges we would like to construct. More importantly, this process is aware of previous constructed edges, both in the current layer and previous layers. The recurrent edge predictor is shown in Figure 1(b). We implement it using a single-layer LSTM model.

Once having constructed all edges for each layer, we can immediately obtain the set of neighbors $\mathcal{N}(e_i^n)$ for each node $e_i$ in the $n$-th layer. Self-attention operations with multi-head mechanism are then performed on its neighbors for each node. For text tasks, we regard each token as a node. For graph-like structures, we treat nodes in the original graph as nodes. For images, the input $(H, W, F_{\text{in}})$ dimensional sensor is reshaped to a $HW \times F_{\text{in}}$ matrix, where each row can be thought as a node by our definition.

### 4.2  Distance Encoding

The input graph intrinsically displays some degree of structures. For example, in a sequence of natural language tokens, the relative distance between two tokens in the sequence or the corresponding parse

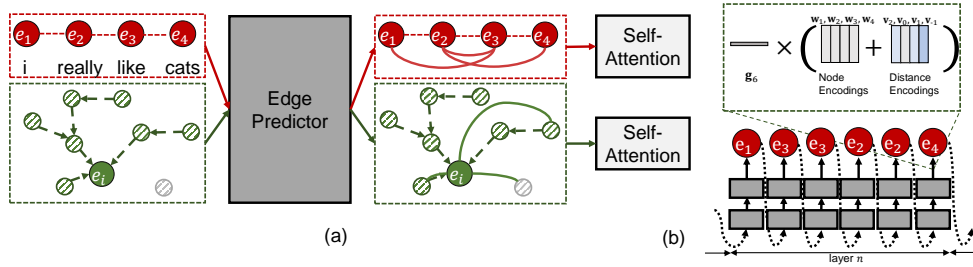

Figure 1: An illustration of the proposed Sparse Apdative Connection. (a) shows the process of SAC to construct edges and then perform self-attention on these edges (Red is for text and green is for graphs). (b) shows the edge prediction process of (a) with distance encodings. When predicting time-step 6, the word embeddings are added with distance encodings.

tree encodes structural information. As another example, in the task of node representation learning in graphs (Yang et al., 2016b; Veličković et al., 2017; Xu et al., 2018), the graph originally comes with the node edges. The LSTM edge predictor described above ignores this structure. To leverage original structural information, we propose distance encodings to incorporate graph structure into the edge predictor. Distance encodings only affect the destination node predictions. In contrast to only using node embedding matrix $\mathbf{W}$, we add an extra *distance matrix* $\mathbf{V}$ that encodes distance information to the original projection matrix $\mathbf{W}$, giving $\mathbf{V} + \mathbf{W}$. Each column in $\mathbf{V}$ is its corresponding distance representation to the current original node.

For example in Figure 1, at time-step 6, when two edges $(e_1, e_3)$, $(e_3, e_2)$, and one origin node $e_2$ have been generated, we are to use $\mathbf{g}_5 \in \mathbb{R}^d$, the output of the LSTM model at time-step 5, to predict the node at time-step 6. According to the original structure, the distance between $e_2$ (the current origin node) and all the nodes $e_1, e_2, e_3, e_4$ by far are 2, 0, 1, and -1 respectively, where -1 means inability to reach. The distance vectors are thus $\mathbf{v}_2, \mathbf{v}_0, \mathbf{v}_1, \mathbf{v}_{-1}$, which are vectors of size $\mathbb{R}^d$ to be learned. Intuitively, this process also discourages generating duplicate edges and leverages the original structural information.

In contrast to Veličković et al. (2017) where attention operations are only performed between nodes with literal edges in the original graph, SAC offers the flexibility in leveraging the original graph structure and influence from the training signals. Additionally, SAC allows for more convenient information exchange between similar nodes that are far away in terms of distance in the original graph structure, which is because the connection construction stage has the ability to connect any pair nodes in the graph. This ability potentially leads to better performances.

### 4.3 Training and Test

Directly training the edge predictor is impractical since we have no access to the ground-truth edges. We use REINFORCE, which is an instance of a broader class of policy gradient methods for optimization. The main idea is to use reinforcement learning to discover the best edge connections for self-attention operations.

Each action $a$ is the node predicted by edge predictor. Let $\boldsymbol{\Theta}$ denote parameters of the edge predictor and $\boldsymbol{\Phi}$ denote the parameters of the main network which maps an input to its final label based on a pre-defined self-attention structure. Under the framework of reinforcement learning, we ask the edge predictor to maximize its reward $\mathcal{R}(\boldsymbol{\Theta})$, which is the log probability of predicting the correct label, e.g., for neural machine translation the reward $\mathcal{R}$ is the average log probability of golden target tokens; for image classification, the reward the log probability of the correct label. Consider the simple case where different attention layers use the same node connections, by sampling a sequence of nodes from the edge predictor, we are able to update the parameters in edge predictor using policy gradients:

$$\nabla J(\boldsymbol{\Theta}) = \sum_{i=1}^{2\alpha N} \nabla \log p(a_i | a_{1:i-1}; \boldsymbol{\Theta})(\mathcal{R}(\boldsymbol{\Theta}) - b) \tag{4}$$

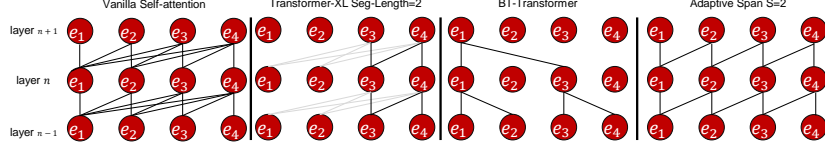

Figure 2: Connection of SAC to other methods for computing self-attention.

where $b$ denotes the baseline which is the average of the previous rewards. $\mathbf{\Phi}$ is updated directly based on the log-likelihood. At test time, edges are decoded using beam search. We use a beam size of 5 for all models.

### 4.4 Variants of Edge Predictor

The vanilla version of the Edge Predictor can be further regulated, simplified or expanded with prior knowledge for preferable graph structures.

**All layers sharing the same structure**  To reduce the computational cost and RL search space, we can enforce the edge structure to be the same for all layers, where the process is only executed once instead of $L$ times. We adopt this strategy for all settings to reduce the search space.

**All nodes connected in each layer**  To enforce each node to be connected in each layer, for each node $e_i$, it is repeatedly fed to the predictor $\alpha$ times as the original node, and we only predict the destination node. The graph can be either directed graph or undirected graph, depending on how we want self-attention to be computed.

**Different heads attending to different contexts (head adaptive for short)**  Sukhbaatar et al. (2019) shows that it is beneficial if different heads attend to different spans (some focusing on the recent history, while others focusing the whole available context). We can also augment the model by assigning each head with a edge predictor, providing the flexibility that different heads can attend to different chunks of context. We sequentially predict all input and output nodes for each head, and the prediction of $2\alpha N$ nodes are repeated $H$ times. In this way, the prediction model for the current head is aware of the information of all previous heads. A head specific embedding is appended to the node embedding in LSTMs to let the model be aware of the current head. Since this strategy significantly increases the search space in RL, we empirically find that it helps some settings, but not always.

### 4.5 Connection to Existing Methods

In this subsection, we describe the connection between SAC and previous variants of self-attentions, and show that these variants computing self-attention can be obtained through SAC if we slightly modify the edge predictor. For ease of exposition, we use $\text{EP}(\mathbf{e}) = \{(e_i, e_j)\}$ to denote the collection of all edges for self-attention operations.

**Connection to vanilla self-attention (Vaswani et al., 2017)**  The vanilla self-attention links each pair of nodes, where $\text{EP}(\mathbf{e})$ can be described as $\text{EP}(\mathbf{e}) = \{(e_i, e_j) | i \in [1, N]\}$.

**Connection to Transformer-XL (Dai et al., 2019)**  Transformer-XL treats the text in a segment-by-segment style. Self-attention operations are performed between nodes within the same segment. $\text{EP}(\mathbf{e})$ can be described as $\text{EP}(\mathbf{e}) = \{(e_i, e_j) | i \in [1, N]; j \in \text{Segment}(i)\}$.

**Connection to Adaptive Span Transformer (Sukhbaatar et al., 2019)**  Adaptive Span Transformer learns an optimal attention span for each head. Suppose the span size assigned to head $t$ is $s$, then $\text{EP}(\mathbf{e}, \mathbf{t})$ can be described by: $\text{EP}(\mathbf{e}) = \{(e_i, e_j) | i \in [1, N]; j = i, i-1, \cdots, i-s+1; \text{span} = t\}$.

**Connection to BP-Transformer (Ye et al., 2019)**  BP-Transformer constructed a tree-like graph by adding span nodes apart from token nodes. There are $2N - 1$ nodes in total, where $N$ is the sequence length. In BP-Transformer, each token (leaf) node attends to each span (non-leaf) node that includes it, which we refer to as Ancestor$(e_i)$ for node $e_i$. It is easy to prove that a leaf node is

| Model | H (heads) | B (blocks) | edges | dev (BLEU) | test (BLEU) | test (cased sacreBLEU) |
|---|---|---|---|---|---|---|
| Transformer Base (Vaswani et al., 2017) | 8 | 6 | $N^2$ | 25.8 | 27.3 | |
| BP base (Ye et al., 2019) | | | | | 28.1 | 27.6 |
| Reversible base (Kitaev et al., 2020) | | | | | 28.0 | 27.4 |
| SAC base | 8 | 6 | $2N$ | 17.4 | 18.3 | 17.8 |
| SAC base | 8 | 6 | $5N$ | 25.6 | 27.0 | 26.2 |
| SAC base | 8 | 6 | $10N$ | 26.0 | 27.7 | 27.0 |
| SAC base | 8 | 6 | $15N$ | 25.6 | 27.4 | 26.8 |
| SAC base | 16 | 6 | $10N$ | 26.2 | 28.1 | 27.6 |
| SAC base | 16 | 12 | $10N$ | 26.4 | **28.4** | 27.8 |
| Transformer big (Vaswani et al., 2017) | 16 | 6 | $N^2$ | 26.4 | 28.4 | |
| Reversible big (Kitaev et al., 2020) | | | | | 29.1 | 28.4 |
| SAC Large | 16 | 6 | $10N$ | 26.7 | 28.9 | 28.1 |
| SAC Large | 16 | 18 | $10N$ | 26.9 | 29.4 | 28.6 |
| SAC Large (dependency) | 16 | 18 | $10N$ | 26.9 | **29.5** | **28.8** |

Table 1: BLEU scores on the newstest2013 for development and newstest2014 for test for WMT English-German. $N$ denotes the length of the input sequence.

associated with $\lfloor \log_2 N \rfloor$ non-leaf nodes (and thus attends to $\lfloor \log_2 N \rfloor$ nodes). Therefore, we have EP(**e**)=$\{(e_i, e_j)|i \in [1, N]; j \in \text{Ancestor}(e_i)\}$.

# 5 Experiments

## 5.1 Machine Translation

We use the encoder-decoder model (Bahdanau et al., 2014; Vaswani et al., 2017) as the backbone for machine translation. For the encoder, SAC constructs $\alpha N$ edges for each layer and self-attention operations are performed between connected nodes. For the decoder, masked attention (Vaswani et al., 2017) is applied. Specifically, given a newly generated target node, it can attend to all source nodes, dummy nodes, target nodes that come beforehand, but not target nodes that come afterwards. We again use SAC to construct edges between the newly generated node and the preceding nodes, where the input node to the edge predictor is forced to be the newly generated node, and the output node is limited to preceding nodes and the dummy nodes.

Following Vaswani et al. (2017); Ott et al. (2018); Kitaev et al. (2020), we used the standard WMT 2014 English-German dataset to test the proposed model. The dataset consists of about 4.5 million sentence pairs. Sentences are encoded using BPE (Sennrich et al., 2016), which has a shared source target vocabulary of about 37000 tokens. For fair comparison, we used the Adam optimizer (Kingma and Ba, 2014) with $\beta_1 = 0.9$, $\beta_2 = 0.98$ and $\epsilon = 10^{-9}$ for all models. Label smoothing (Szegedy et al., 2016) with $\epsilon = 0.1$ is applied for all models. For the base setup, following Vaswani et al. (2017), the dimensionality of inputs and outputs $d_{\text{model}}$ is set to 512, and the inner-layer has dimensionality $d_{\text{ff}}$ is set to 2,048. For big models, $d_{\text{model}}$ is set to 1,024 and $d_{\text{ff}}$ is set to 4,096. Models are run on 8 NVIDIA V100 GPUs.

Results are shown in Table 1. As we gradually increase the number of edges for each layer (from 2 to 5 to 10 to 15 per node), we can see that the performance first increases, reaching the highest with $\alpha$ set to 10, and then decreases. This means that performing attention operations between all pairs is not only unnecessary, but can hurt the performance. Memory saved from sparse connections allow for more heads to perform attentions and deeper networks with more blocks, leading to better performances over vanilla transformers. We also implement a dependency-based model, in which English sources were first parsed using Stanford Dependency parser (Chen and Manning, 2014). Relative positions between nodes in the dependency trees are encoded in distance encodings of the edge predictor. The introduction of dependency parser for attention construction introduces +0.14 BLEU score boost. We did not observe significant performance boost from the head-adaptive strategy, and thus omit their performances.

| Method | Enwiki8 | Text8 | Params |
|--------|---------|-------|--------|
| Trans (Al-Rfou et al., 2019) | 1.11 | 1.18 | 44M |
| Trans-XL (Dai et al., 2019) | 1.06 | - | 41M |
| Adaptive(Sukhbaatar et al., 2019) | 1.02 | 1.11 | 39M |
| BPT (Ye et al., 2019) | 1.02 | 1.11 | 38M |
| SAC (basic) | 1.02 | 1.07 | 39M |
| SAC (head adaptive) | **1.00** | **1.06** | 39M |

Table 2: Performances on language modeling datasets.

## 5.2 Language Modeling

We use character-level language modeling datasets to evaluate SAC's ability to handle long-term dependencies. We use Enwiki8 (Mahoney, 2011) and Text8 (Mahoney, 2011) for evaluation and report the values of BPC for different models. We use the Transformer decoder architecture as the backbone. We compare SAC with other variations of transformers to fit long sequences into the model, including the vanilla Transformer (Al-Rfou et al., 2019), which splits the whole sequence into smaller segments, and only trains the model within each segment and ignore the rest; Transformer-XL (Dai et al., 2019) that adopts a recurrence mechanism to cache the memory of previous segments; adaptive span model (Sukhbaatar et al., 2019) that assigns different heads with different text spans in an adaptive fashion; and the BP-Transformer (Ye et al., 2019) that splits the sequence using binary trees. For SAC, $\alpha$ is set to 256 for each node. The relatively small memory cost allows the model to look at a maximum context of 50k characters. Input dimensionality is set to 512, and the inner-layer dimensionality 2,048. Following (Sukhbaatar et al., 2019), we use Adagrad for optimization, with a batch size of 64 and fixed learning rate of 0.07 and 32k warm-up steps.

Results are shown in Table2. As can be seen, SAC-basic outperforms the other Transformers by 0.04 bcp on Text8 while significantly reducing the memory usage for large attention spans. For Enwiki8, it ties with the best BPT model, achieving 1.02 bcp score. The improvement validates the importance modeling long-term dependencies with limited available memory. We also find that, in the language modeling tasks, the head-adaptive strategy helps,

## 5.3 Representation Learning in Graphs

We test the performance of the proposed model on both transductive and inductive benchmark datasets. For the transductive setup, we used the three standard citation network benchmarks, Cora, Citeseer and Pubmed (Sen et al., 2008). In the transductive setup, we used the Protein-protein interaction dataset (PPI) (Zitnik and Leskovec, 2017). The training algorithm has access to all of the nodes' feature vectors and labels, and predictions are performed on the test nodes. The detailed descriptions for Cora, Citeseer, Pubmed and PPI are found in the Appendix due to the space limit.

The difference between SAC and (Veličković et al., 2017) is that the latter performs self-attention operations between nodes that are connected though graph edges, while SAC perform self-attention operations between nodes linked by the edge predictor. For fast convergence, we initialize SAC using the pretrained attention model (Veličković et al., 2017), where attention links are just edges in the original graph. Then we start exploring edge construction across all nodes. the number of attention heads is fixed to 8 and the number of blocks is set to 12. We experiment different values of $\alpha$, i.e, [5, 10, 50, 100] unless the memory usage reaches limitation.

We train all models with Adam (Kingma and Ba, 2014) and early stopping on the validation set. The initial learning rate is treated as a hyper-parameter trained on the validation set. Following (Veličković et al., 2017), we run 100 epochs in total and use an early stopping strategy on the both the cross-entropy loss and accuracy for transductive tasks and micro-F1 for inductive tasks. Each experiment is repeated three times and we report the mean value.

Results are shown in Table 3. We note that SAC achieves significant performance boosts over existing methods across all four datasets, i.e., outperforms our implemented GAT +1.8, +1.1, +0.7 and +1.1 respectively on Cora, Citeseer, Pubmed and PPI. The explanation for SAC's advantage is as follows: graph node representation learning concerns about both label propagation and relatedness between nearby nodes in the vector space, the latter of which is what GCN handles. As verified in many

| Available data | Method | Cora | Citeseer | Pubmed | PPI |
|---|---|---|---|---|---|
| **A** | DeepWalk (Perozzi et al., 2014) | 67.2 | 43.2 | 65.3 | – |
| **X, A** | DGI (Veličković et al., 2019) | 82.3 | 71.8 | 76.8 | 63.8 |
| **X, A** | GraphSAGE (Hamilton et al., 2017a) | – | – | – | 50.2 |
| **X, A, Y** | SemiEmb (Weston et al., 2012) | 59.0 | 59.6 | 71.7 | – |
| **X, A, Y** | Planetoid (Yang et al., 2016a) | 75.7 | 64.7 | 77.2 | – |
| **X, A, Y** | Chebyshev (Defferrard et al., 2016) | 81.2 | 69.8 | 74.4 | – |
| **X, A, Y** | GCN (Kipf and Welling, 2016) | 81.5 | 70.3 | 70.0 | – |
| **X, A, Y** | MoNet (Monti et al., 2017) | 81.7 | – | 78.8 | – |
| **X, A, Y** | SGC (Wu et al., 2019) | 81.0 | 71.9 | 78.9 | – |
| **X, A, Y** | AdaLNet (Liao et al., 2019) | 80.4 | 68.7 | 78.1 | – |
| **X, A, Y** | SGAT (Ye and Ji, 2019) | 84.2 | 68.2 | 77.6 | 96.6 |
| **X, A, Y** | CurvGN-n (Ye et al., 2020) | 82.7 | 72.1 | 79.2 | – |
| **X, A, Y** | GAT (Veličković et al., 2017) | 83.0 | 72.5 | 79.0 | 97.3 |
| **X, A, Y** | SAC | **84.8** | 73.8 | 79.7 | **98.4** |
| **X, A, Y** | SAC (head adaptive) | 84.7 | **74.0** | **80.1** | **98.4** |

Table 3: Summary of results in terms of classification accuracies on transductive tasks (Cora, Citeseer and Pubmed) or micro-averaged F1 score on inductive tasks (PPI). In the first column, we report the kind of data available to each method during training (**X**: features, **A** adjacency matrix, **Y**: labels).

| | CIFAR100 | | | | | ImageNet | | | |
|---|---|---|---|---|---|---|---|---|---|
| | GFlops | top1 | top5 | Params | | GFlops | top1 | top5 | Params |
| WideResNet | 10.4 | 80.3 | 95.0 | 36.3M | ResNet50 | 8.2 | 76.4 | 93.1 | 25.6M |
| Bello et al. (2019) | 10.9 | 81.6 | 95.2 | 36.2M | | 8.3 | 77.7 | 93.8 | 25.8M |
| SAC | 11.0 | 82.2 | 95.4 | 36.2M | | 8.3 | 78.5 | 94.2 | 25.9M |
| SAC (head adaptive) | 11.0 | **82.4** | **95.5** | 36.2M | | 8.3 | **78.7** | **94.3** | 25.9M |

Table 4: Results of image classification on CIFAR-100 using the Wide-ResNet 28-10 Zagoruyko and Komodakis (2016) as the backbone and on ImageNet using the ResNet-50 He et al. (2016) model.

recent works Liu et al. (2018); Wang and Leskovec (2020), combining both facets leads to better performances. The attention edge prediction stage in SAC fosters information exchange between nodes that are not directly linked in graph but similar in terms of label propagation. SAC actually offers the probability in bridging the aspects, leading to better performances.

## 5.4 Image Classification

Augmenting convolution models with self-attention (Bello et al., 2019; Parmar et al., 2019; Hu et al., 2019; Wang et al., 2019) provides the model with the ability to capture global contexts in an image and has yielded gains in several vision tasks such as image classification and objective detection. We follow the protocols in (Bello et al., 2019), i.e. incorporating relative position embeddings for self-attention operations and augmenting each ResNet (Zagoruyko and Komodakis, 2016; He et al., 2016) block with self-attentions. To handle the prohibitive memory cost, (Bello et al., 2019) performs self-attention operations starting from the last layer, which has the smallest spatial dimension, until memory constraints are hit. This ad-hoc strategy is replaced by SAC. Following (Bello et al., 2019), we conduct experiments on CIFAR-100 (Krizhevsky et al., 2009) and ImageNet (Deng et al., 2009). For CIFAR-100, we use the Wide-ResNet-28-10, the architecture of which comprises 3 stages of 4 residual blocks each using two $3\times3$ convolutions. We augment each convolution of all residual blocks with the number of attention heads set to 16. For ImageNet, we use ResNet-50, the block of which consists of $1\times1$, $3\times3$, $1\times1$ convolutions where the last pointwise convolution expands the number of filters and the first one contracts the number of filters. We tune $\alpha$ in range $\{5, 10, 20\}$.

Results are shown in Table 4. As can be seen, the proposed SAC model significantly outperforms the attention model in (Bello et al., 2019) with the only modification of automatic edge construction. Specifically, the top-1 score increases from 81.6 to 82.4 for CIFAR-100 and from 77.7 to 78.7 for ImageNet. The improvement validates the importance of performing necessary attention operations under memory limit.

# 6  Conclusion

In this work, we propose Sparse Adaptive Connection — a sparse connection method to accelerate and structure the self-attention mechanism that adapts to various downstream tasks. We use an LSTM edge predictor to construct edges for self-attention operations, which gives us control of how sparse we want self-attention to be by setting the sparse coefficient $\alpha$. We demonstrate that SAC is competitive with state-of-the-art models on neural machine translation, language modeling, graph classification and image classification, while reducing memory costs.

## Broader Impact

Accelerating fully-connected self-attention has been a research trend in recent years. Vanilla self-attention models, such as Transformers and BERT, are not able to process extremely long text, where text must be in advance segmented into pieces and then can be individually modelled. The lack of adequate context leads to poor performances in generating long, coherent and fluent text. The goal of our proposed method, SAC, is to provide a way of relieving the computation burden of vanilla self-attention by automatically searching for the best attention patterns. We believe SAC has great potentials to generate high-quality long text. While there is risk of abuse, like generating fake news, the value of SAC is generally safe and weighs more than abuse to the whole society.

## Acknowledgement

We thank all reviewers for their insightful comments. We also want to thank Zihao Ye for his helpful suggestions on evaluations, along with suggestions on learning head-specific policies.

## Footnotes

[1] We use the term "node' in a broad sense of denoting any particular unit in text, images or graphs.

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
