[Supplementary Material]

# Supplementary Materials for
# SAC: Accelerating and Structuring Self-Attention via Sparse Adaptive Connection

## 1 Datasets

For the transductive setup, we used the three standard citation network benchmarks, Cora, Citeseer and Pubmed (Sen et al., 2008). We followed the transductive setup adopted in (Yang et al., 2016; Veličković et al., 2017; Xu et al., 2018), where nodes correspond to documents and edges to (undirected) citations. Cora contains 2708 nodes, 5429 edges, 7 classes and 1433 features per node. Citeseer contains 3327 nodes, 4732 edges, 6 classes and 3703 features per node. Pubmed contains 19717 nodes, 44338 edges, 3 classes and 500 features per node.

For the inductive setup, we used the Protein-protein interaction dataset (PPI) (Zitnik and Leskovec, 2017), which aims at classifying protein roles such as cellular functions and gene ontology in various protein-protein interaction (PPI) graphs, where each graph corresponds to a different human tissue. Critically, testing graphs remain completely unobserved during training. The dataset has 56.9K nodes, 806.2 edges with 121 classes. The average number of nodes per graph is 2372. Each node has 50 features that are composed of positional gene sets, motif gene sets and immunological signatures. There are 121 labels for each node set from gene ontology, collected from the Molecular Signatures Database (Liberzon et al., 2011), and a node can have several labels simultaneously.

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

## References

Arthur Liberzon, Aravind Subramanian, Reid Pinchback, Helga Thorvaldsdóttir, Pablo Tamayo, and Jill P Mesirov. 2011. Molecular signatures database (msigdb) 3.0. *Bioinformatics*, 27(12):1739–1740.

Figure 1: Predicted self-attention links for the text *barack obama is an american politician and attorney who served as the 44th president of the president of the united states from 2009 to 2017. as a member of the democratic party, he was the first african-american president of the united states. he previously served as a u.s. senator from illinois from 2005 to 2008 and an illinois state senator from 1997 to 2004.*

Prithviraj Sen, Galileo Namata, Mustafa Bilgic, Lise Getoor, Brian Galligher, and Tina Eliassi-Rad. 2008. Collective classification in network data. *AI magazine*, 29(3):93–93.

Petar Veličković, Guillem Cucurull, Arantxa Casanova, Adriana Romero, Pietro Lio, and Yoshua Bengio. 2017. Graph attention networks. *arXiv preprint arXiv:1710.10903*.

Keyulu Xu, Chengtao Li, Yonglong Tian, Tomohiro Sonobe, Ken-ichi Kawarabayashi, and Stefanie Jegelka. 2018. Representation learning on graphs with jumping knowledge networks. *arXiv preprint arXiv:1806.03536*.

Zhilin Yang, William W Cohen, and Ruslan Salakhutdinov. 2016. Revisiting semi-supervised learning with graph embeddings. *arXiv preprint arXiv:1603.08861*.

Marinka Zitnik and Jure Leskovec. 2017. Predicting multicellular function through multi-layer tissue networks. *Bioinformatics*, 33(14):i190–i198.