[Reviews · NeurIPS 2020]

Review 1

Summary and Contributions: This work proposes a new way to apply the self-attention mechanism. Instead of naively applying the quadratic complexity self-attention among a set of objects (either nodes of a graph or words in a sentence) they propose to prune some of the links via a learnable function. The authors demonstrate that their model is competitive with state-of-the-art models while significantly reducing memory cost.

Strengths: * well connected to the recent literature; * the method is explained clearly both theoretically and with technical details; * the experiments are well motivated, well-described and they prove the right points.

Weaknesses: * improvements wrt other methods are marginal;

Correctness: Claims are correctly motivated both referring to current literature and with empirical evidence.

Clarity: The paper is well written. All points are made clearly and the math/ notation is consistent along with the paper and with the current literature.

Relation to Prior Work: Yes it is, there is a specific section that connect this work to previous contributions and connections are well discussed.

Reproducibility: Yes

Additional Feedback:


Review 2

Summary and Contributions: The paper proposed a mechanism for sparsifying self-attention in Transformers for efficiency. Before each self-attention sublayer, an RNN generates selections dictating which tokens can attend to which tokens. This generation process is trained by RL together with the supervised training of the main model. The experiments are done on NMT, LM, graph tasks and image classification.

Strengths: The paper is tackling an important problem of reducing the computational cost of Transformers. As Transformers are being used everywhere, making it efficient obviously has a large impact. The proposed approach is interesting and novel, and quite different from most sparse attention methods. Actually, it's really surprising that such large-scale discrete optimization can be solved with RL, especially together with a supervised optimization of a large model. The experiments are done on sufficiently large-scale real tasks such as enwik8, where the model is working well.

Weaknesses: My main concern is about the computational cost the proposed method. The method requires running a LSTM on each token on every layer (or even every head) sequentially. Compared to the parallel processing of Transformers, I would expect this sequential computation to be quite slow. (i) From the Text8 experiment numbers, my rough calculation is 512(nodes)x256(edge-per-node)x12(assumed layers)x8(assumed heads)=12M steps of a LSTM all run sequentially. (ii) In addition, the output layer of LSTM has a size that grows with the input sequence length, and it's not shared between samples in a batch like it is a normal LSTM. (iii) Even after the sparse connections are found by the LSTMs, computing a sparse attention requires a sparse matrix multiplication (MM), which is not as efficient as a dense MM used by self-attention. All those factors should affect the computation speed in a negative way. Given that the computational efficiency is the goal of the paper, the authors must discuss them in detail. The paper fails to mention the actual numbers regarding the saved computation in the experiments. In L222, it's not mentioned how long the trained actually took and how much it was faster. The claimed memory reductions in L226, L249 are never explicitly quantified. Without those numbers, it is hard to judge the gain brought by the proposed method.

Correctness: I think it's not correct to claim that the other sparse attention methods are special cases of the proposed method SAC. In sec 4.5, 4 methods described as having a specific sparse connection scheme. But this doesn't mean they're special cases of SAC. Using RL trained RNNs to find sparse connections is the fundamental part of SAC, but none of those 4 paper has anything similar to that. Other minor comments: - The proposed method make use of a beam search during testing as mentioned in L183. Even though it's used for the edge selection and not output selection, it's still not fair to compare with baselines without any beam search. - I'm bit surprised by the discrepancy between the Enwik8 and Text8 experiments. In my experience, if a method is better on one of them they usually work better on the other too, which is expected as those datasets are constructed from the same source. Here we see 0.04bpc difference in one and none in the other.

Clarity: Yes, the paper is written clearly. I just have few minor comments: - Lines 21-22 are giving an impression that Transformer-XL has a quadratic complexity, which is not true. Transformer-XL has a fixed attention span, so it has a linear complexity with regards to the sequence length. - L44 mentions "recurrence" in Transformer-XL, which is misleading as Transformer-XL is strictly a feedforward model. Also, relative position encoding is proposed before that in [1,2]. - L94: It might be better use L as the number of layers, and N as the number of tokens, not the other way around. - There might be a typo in L105 as "representation" is mentioned twice. - L102: It is never said what is the ordering this set? I inferred from the rest of the paper that it is ordered as {node A of edge 1, node B of edge 1, node A of edge 2, node B of edge 2,..}. - L224: probably a typo 20 -> 15. - L241: wrong reference. - L246: typo [1]: Shaw et. al. "Self-Attention with Relative Position Representations", 2018 [2]: Sukhbaatar et. al. "End-to-end memory networks", 2015

Relation to Prior Work: See the first paragraph the correctness section.

Reproducibility: Yes

Additional Feedback: The paper can be stronger if the above mentioned concerns about the computational efficiency are addressed. == Post-rebuttal comment == I'm keeping my score as it is because my concern about the actual efficiency of the method wasn't sufficiently addressed in the rebuttal.


Review 3

Summary and Contributions: The authors propose an Edge Predictor which utilizes an LSTM model to dynamically build the self-attention alignments. The proposed method is competitive with state-of-the-art models on WMT, LM and Image classification tasks while significantly reducing memory cost. Overall, although the method looks interesting and the performance looks good, sequential edge by edge prediction with LSTM may not draw much attention in community. When sequences are long, it will also take long time. When sequences are short, self-attention can be computed in parallel. Some other ways, such as Reformer which can build sparse attention in parallel, would be more interesting. Moreover, the authors only report the accuracy of SAC, but missing the results on speed to support the title "Accelerating Self-Attention". [1] Reformer: The Efficient Transformer

Strengths: 1. The proposed method can dynamically build the hard attention in text sequence or image by making use of reinforcement learning. 2. The proposed method can be widely adopted in many different tasks.

Weaknesses: 1. The proposed method may not be able to accelerate the whole framework with GPU, as the edge prediction can not be computed in parallel. The title may need to modify, otherwise more experiment results on speed should be provided to valid the statement "accelerating self-attention". 2. Need more detailed analysis on how much memory can be saved compared to other methods, such as XLNet or reversible Transformer/CNN (REFORMER: THE EFFICIENT TRANSFORMER). The explored datasets actually do not need huge memory. It would be better to explore some tasks with much longer sequences. 3. The number of parameters in Table 1 should be listed for fair comparison. 4. No experiment result or ablation study on "Variants of Edge Predictor". It's unclear which method plays the key role.

Correctness: Yes

Clarity: Yes

Relation to Prior Work: Yes

Reproducibility: Yes

Additional Feedback:


Review 4

Summary and Contributions: The paper proposes a Sparse Adaptive Connection (SAC) model based on Transformer. SAC learns to predict sparse connections (attention links) between inputs and attentions are only performed on those predictive links. Thus, SAC replaces the dense attention matrix in the standard Transformer. Without ground-truth links to predict, SAC are trained with REINFORCE with log-probability of the output as the reward. The authors evaluate SAC on four tasks (machine translation, language modeling, graph prediction, image classification) and show that SAC performs better than previous works.

Strengths: - The paper proposes an interesting approach for modeling sparse attention in Transformer using another neural network (LSTM) to predict which nodes should attend to which nodes. The method allows for controlling the number of links per layer in the Transformer. - Evaluation on a different tasks and datasets. This shows the proposed method is applicable for a while range of applications.

Weaknesses: - One particular weakness of the propose method is the use of the LSTM predictor. this sequential predictor can slow the training and inference of the model. Especially during the inference, a beam search run first to decode the set of links between nodes. The training time and inference time should be reported in the paper to highlight the trade off between performance and latency. For language modeling task, where \alpha is large (256) and L is large (50K), beam search might be very slow. The use of LSTM predictor might be problematic for the proposed approach when the input sequence is getting longer (i.e modeling document level). - While the authors compare SAC to previous work, I wonder what is the advantage of the LSTM link predictor compared to manual design links between nodes. For language modeling and machine translation tasks, local n-gram are strong features for the task (this is also why CNN models in fair-seq do well in NLP). A baseline that uses \alpha links per node could be the model where each node links to \alpha nearest nodes (or \alpha previous nodes). Such a baseline is important in my opinion to highlight advantage of SAC where the prediction of links might be very slow.

Correctness: While the results show that SAC performed better than previous work. I find the analysis of why this is the case is missing. It has been shown that dense attention might hurt the performance (Fixed Encoder Self-Attention Patterns in Transformer-Based Machine Translation Alessandro Raganato, Yves Scherrer, Jörg Tiedemann) but what kind of pattern that SAC discovered is useful? For instance, by looking at the Supplementary I saw that attention links often connect similar words.

Clarity: The details of the LSTM predictor could be made more explicit (e.g how to initialize the LSTM in the higher layer of the transformer). Maybe explicit equations could help to understand and implement the idea.

Relation to Prior Work: A very similar work is missing in the citation. Adaptively Sparse Transformers. Correia et al, 2019. This work tackle the same problem and proposed a method to learn a sparse connection between nodes per layer (without predefined parameter \alpha). Adaptive Sparse Transformers do not rely on a recurrent predictor, thus it fast. I find that SAC should be compared with Adaptively Sparse Transformers.

Reproducibility: No

Additional Feedback: - What is the training/decoding speed of SAC? == Post-rebuttal comment == I keep my score after reading the author response.

[Author Response · NeurIPS 2020]

**General Response**   We thank all reviewers for their insightful comments! We are sorry that Figure 1 in the submission version of this paper is not the lastest so the descriptions in Section 4.3 are ambiguous. We will correct it in the updated version. We have revised typos and removed duplicated sentences.

**Regarding computation efficiency and memory cost.**   For memory, SAC consumes smaller memory since the connections are sparse. For computation efficiency, at training time, SAC is only slightly faster than vanilla transformers, since learning which node should be linked to which node using RL is time-consuming. But at test time, SAC is significantly faster due to significantly smaller cost in self-attention computations. We will add more details in the updated version.

**To Reviewer #1**   Yes, we agree that the improvements compared with state-of-the-art models are marginal. But the main goal of this paper is to **reduce the memory cost of vanilla self-attention while achieving slightly better performances**. We do not attempt to improve the results, and instead, we show that with less attention connections, the model is also strong and consistent over tasks.

**To Reviewer #2**   Thanks for your careful and insightful feedback! We will correct all the typos and incorrect references in the next version. For the order, it is indeed what you think, i.e. $\{a_{11}, a_{12}, a_{21}, a_{22}, \cdots\}$, where $a_{i1}$ is the start node for the $i$-th edge and $a_{i2}$ is the end node.

`Correctness`: Yes, we cannot directly say the other four methods are "special cases" of SAC, but by imposing extra constraints when training the LSTM edge predictor, we can actually induce each of them. For example, for vanilla self-attention (take the encoder side as an example), we can feed each node $N$ times into the predictor so that it recovers vanilla self-attention with the help of distant encodings. For Transformer-XL, we can still segment the sequence and apply the above operations.

In terms of decoding in MT, all models (including baselines) use beam search. We are sorry for the confusion and will make this point clearer in the updated version.

`Weakness`: See **General Response**.

**To Reviewer #3**   Thanks for you helpful and insightful comments!

`Weakness 1&2`: See **General Response**.

`Weakness 3`: We omit the parameters due to the limited space of the page. In fact, the numbers of parameters for these methods including SAC are very close, as you can see from Table 2 and Table 4 that the most parameters come from the main model, i.e. $\Phi$ rather than $\Theta$.

`Weakness 4`: Thank you for the sensible comment. We used a simple version of the edge predictor, in which all layers share the same structure and each node has to be connected to some other nodes for each layer. For the head adaptive strategy, we reported results for both (head adaptive or not adaptive) for different tasks. We will make these points clearer in the updated version.

**To Reviewer #4**   We thank you for your insightful comments!

`Weakness 1`: See **General Response**.

`Weakness 2`: Yes, at first glance, the inference speed of SAC is slower than vanillan Transformer since it introduces the extra process of link prediction. But in fact, SAC does not need to do full self-attention, which makes a significant remedy for test speed. We will show these in the next version. As for baselines, connecting to nearest nodes is actually what CNNs do, for which many recent works have discovered for sparse self-attention. We will compare the performance and the speed of these methods.

`Correctness`: Thanks for your suggestions! The intuition of using LSTM edge predictor is to learn different attention patterns for different downstream tasks. In NMT, we find that the learned attention prefers more semantics-related words. We will give a deeper analysis and plot more figures to show how SAC construct attentions for different tasks.

`Clarity`: The LSTM predictor is initialized randomly (uniform and guassian distributions are both possible). We will clarify this in the next version.

`Reference`: We are sorry for the confusion and will make it clearer in the updatedd version.

[Meta-Review · NeurIPS 2020]

This paper addresses the quadratic bottleneck in transformer architecture. It proposes a Sparse Adaptive Connection (SAC) model which learns to predict sparse connections (attention links) between inputs and attentions are only performed on those predictive links. The proposed method is competitive with state-of-the-art models on WMT, LM and Image classification tasks while significantly reducing memory cost. Overall, three of the four reviewers seem to have liked the paper, although they had some concerns (below), while one reviewer (R3) proposed weak reject. A weakness pointed out by R2 and R3 is that only accuracy is reported, but speed is not, which seems necessary to support the title "Accelerating Self-Attention". The authors promised to add more details about computational efficiency and memory cost in the final version, and I urge them to do so. While I am a bit in the fence with this paper, I still think the proposed approach is interesting and therefore recommend acceptance. However, I strongly urge the authors to follow the reviewers’ recommendations to improve the paper. Besides the weakness mentioned above, two reviewers (R2 and R4) pointed out missing related work, in particular R4 says a comparison against Correia et al. (2019)’s “Adaptively Sparse Transformers” is needed. This is indeed an important piece of related work that is missing and a discussion should be added in the context of the proposed approach. While a empirical comparison would indeed be nice, I don’t think it is crucial, given that the goal of Correia et al. is not to accelerate transformers (e.g., their method still takes quadratic time). I urge the authors to cite this work along to the two citations suggested by R2 in the context of relative position encoding.